# Nonenzymatic Lactic Acid Detection Using Cobalt Polyphthalocyanine/Carboxylated Multiwalled Carbon Nanotube Nanocomposites Modified Sensor

**Wenqing Shao †, Jiayu Mai † and Zhenbo Wei ***

Department of Biosystems Engineering, Zhejiang University, 866 Yuhangtang Road, Hangzhou 310058, China; 21913051@zju.edu.cn (W.S.); 12013022@zju.edu.cn (J.M.)
* Correspondence: weizhb@zju.edu.cn; Tel.: +86-0571-88982178
† These authors contributed equally to this work.

**Abstract:** In this study, a novel cobalt polyphthalocyanine/carboxylic acid functionalized multiwalled carbon nanotube nanocomposite (CoPPc/MWCNTs-COOH) to detect lactic acid was successfully fabricated. The nanocomposite was systematically characterized by scanning electron microscopy, Fourier transform infrared spectroscopy, ultraviolet–visible absorption spectroscopy, and X-ray photoelectron spectroscopy. The nanocomposite provided excellent conductivity for effective charge transfer and avoided the agglomeration of MWCNTs-COOH. The electrochemical surface area, diffusion coefficient and electron transfer resistance of the CoPPc/MWCNTs-COOH glassy carbon electrode (CoPPc/MWCNTs-COOH/GCE) were calculated as $A = 0.49$ cm$^2$, $D = 9.22 \times 10^{-5}$ cm$^2$/s, and $R_{ct} = 200\ \Omega$, respectively. The lactic acid sensing performance of the CoPPc/MWCNTs-COOH was evaluated using cyclic voltammetry in 0.1 M PBS (pH 4). The results demonstrated that the novel electrode exhibited excellent electrochemical performance toward lactic acid reduction over a wide concentration range (10 to 240 μM), with a low detection limit (2 μM (S/N = 3)), and a reasonable selectivity against various interferents (ascorbic acid, uric acid, dopamine, sodium chloride, glucose, and hydrogen peroxide). Additionally, the electrode was also successfully applied to quantify lactic acid in rice wine samples, showing great promise for rapid monitoring applications.

**Keywords:** nanocomposites; lactic acid; cobalt phthalocyanine; carbon nanotube; nonenzyme

## 1. Introduction

Chinese rice wine, as a traditional alcoholic beverage with high nutritional value and a unique flavor, has been popular in Eastern Asia for centuries. Lactic acid plays an important role in influencing the flavor of Chinese rice wine and as a precursor in the Maillard reaction. Lactic acid can effectively enhance the richness of umami, extend the aftertaste of freshness, balance out the sweetness, and modify the flavor of the wine through synergistic effects with other compounds. Rice wine is made from glutinous rice through a series of processes (such as soaking, steaming, stirring, fermentation, squeezing, ageing, and blending); the lactic acid concentration changes primarily in three of these processes, namely, soaking, fermentation and aging. During soaking, rice swells and the main components are digested with increasing water acidity. During fermentation, saccharides are converted into lactic acid by the metabolic activities of lactic acid bacteria. During the aging process, lactic acid is involved in esterification and influences the development of the wine bouquet. Therefore, the lactic acid concentration should be monitored for quality control during the different stages of wine brewing.

A variety of analytical methods are available for monitoring lactic acid levels in beverages, meat, milk, etc., such as colorimetry, spectrophotometry, titration, chromatography, and proton nuclear magnetic resonance spectroscopy [1–6]. However, sample preparations for these methods are tedious and time consuming, and some methods also require

skilled personnel and the use of expensive equipment. In recent years, biosensing methods utilizing lactate oxidase and lactate dehydrogenase have been studied. Amperometric biosensing is considered a rapid and inexpensive lactate-recognition technique [7–10]. However, because of the inherent fragility of enzymes, the enzyme-based sensors used in this approach inevitably suffer from long-term stability issues due to their sensitivity to harsh environments (pH, temperature, etc.). Another technical drawback is the requirement of rigorous operating conditions and complicated techniques for immobilizing enzymes, which strongly affect their reproducibility and practical applications.

Compared with enzymatic sensors, enzyme-free electrochemical sensors based on nanotechnology have the advantages of simplicity, reproducibility, and good long-term stability in aggressive environments [11–14]. Carbon nanotubes (CNTs) have attracted considerable attention as electrode modifying materials owing to their large surface area, high electron transfer, and excellent chemical stability [15–17]. However, CNTs tend to aggregate in aqueous environments because of the strong $\pi$–$\pi$ interactions and van der Waals forces, which limits their electrochemical properties and can reduce their selectivity to target molecules when used in sensor devices [18–20].

Conducting polymers (CPs), also called conjugated polymers, are a class of functional polymers with $\pi$- electrons that are delocalized over the polymeric backbone [21–24]. These CPs can be deposited inside the walls of carbon nanotubes to obtain a nanocomposite film with higher electrical conductivity and mechanical strength. These hybrid nanocomposite films exhibit attractive properties such as large surface area, chemical stability, and good biocompatibility [25–27]. Moreover, the combination of CPs and CNTs can prevent the agglomeration of CNTs by steric hindrance and electrostatic interactions. The nanocomposite used as a modifying material can enhance the selectivity of the electrode by providing more chemically active sites for the target substances [28–31]. Although CPs/CNTs nanocomposite-modified electrodes have proven to be a potential method for sensing tyrosine, dopamine, glucose, ascorbic acid, and other organic substances, the fabrication of reliable and selective L-lactic acid sensors based on CPs/CNTs nanocomposites has not been reported.

In this study, cobalt phthalocyanine (CoPc) and carboxylic acid functionalized multi-walled carbon nanotubes (MWCNTs-COOH) were used to fabricate a poly-CoPc/MWCNTs-COOH (CoPPc/MWCNTs-COOH) nanocomposite modified lactic acid sensor. CoPc, which has a two-dimensional 18-$\pi$-electron aromatic system, consists of an aromatic organic macrocyclic molecule centered around a single cobalt atom [32,33]. CoPc has rich redox chemistry with high electron-transfer ability, and has been widely used to fabricate sensors because of its excellent electrocatalytic activity for many compounds [34–36]. MWCNTs consist of several nanotubes (with diameters ranging from 5 to 50 nm) composed of graphene sheets nested within one another, and they are widely recognized as the optimal skeleton for modification [37–40]. MWCNTs-COOH are easily dispersed in organic solvents, which improves the homogeneity of the MWCNTs-COOH within the nanocomposites. Therefore, a cobalt polyphthalocyanine (CoPPc)/MWCNTs-COOH nanocomposite-modified electrode was first fabricated as a new electrochemical L-lactic acid sensing platform. The aims of this study are to establish suitable electrodeposition conditions for the fabrication of a CoPPc/MWCNTs-COOH nanocomposite and to investigate the electrochemical behavior and redox mechanism of L-lactic acid by cyclic voltammetry (CV) and differential pulse voltammetry (DPV).

## 2. Materials and Methods

### 2.1. Materials

CoPc, MWCNTs, disodium hydrogen phosphate (Na$_2$HPO$_4$) and sodium dihydrogen phosphate (NaH$_2$PO$_4$) were obtained from Aladdin Chemical Co., Ltd. (Shanghai, China). 0.2 M Na$_2$HPO$_4$·12H$_2$O and NaH$_2$PO$_4$·2H$_2$O were mixed to prepare Phosphate buffer saline (0.1 M PBS, with pH ranged from 4 to 8). HNO$_3$ (65.0–68.0%) and H$_2$SO$_4$ (95.0–98.0%) were purchased from Sinopharm Chemical Reagent Co., Ltd. (Shanghai, China). L-lactic acid and

Nafion were bought from Sigma-Aldrich Co., Ltd. (Shanghai, China). All chemicals were of analytical grade, and all aqueous solutions throughout the experiments were prepared with ultra-pure water (18.25 M$\Omega$ cm$^{-1}$).

### 2.2. Fabrication of CoPPc/MWCNTs-COOH Nanocomposites Modified Sensor
#### 2.2.1. Functionalization of MWCNTs

The morphology of the commercial MWCNTs (MWCNTs were >95% pure, 8–15 nm in diameter, and about 50 μm in length) was examined by transmission electron microscopy (TEM) (Figure 1a). The intrinsic electrical and mechanical properties, and biological activities of MWCNTs were jeopardized because the long MWCNTs intertwined with each other and contained many impurities (such as metal catalyst particles and amorphous carbon fragments). Surface functional groups can alter the functionality and reactivity of MWCNT surfaces and improve their dispersibility and biocompatibility. In this study, we carried out the carboxylation of commercial MWCNTs as follows:

1.  Purification of MWCNTs. Initially, 5 mg MWCNTs was ultrasonically dispersed in 75 mL of $H_2SO_4$ at 25 °C for 30 min. Secondly, the suspension liquid was poured into a 250-mL conical flask and refluxed at 90 °C for 3 h with constant stirring. Then, the mixture was centrifuged at 6000 rpm and washed with deionized water several times until the pH was nearly neutral. Finally, the purified MWCNTs were obtained after drying in a vacuum oven at 70 °C. Impurities were removed by $O_2$, gas or soluble nitrate. The reaction equations were as follows:

$$3C + 4HNO_3 = 3CO_2 \uparrow + 4NO + 2H_2O \tag{1}$$

$$Ni + 4HNO_3 = Ni(NO_3)_2 + 2NO_2 \uparrow + 2H_2O \tag{2}$$

$$Ni + 4HNO_3 = 2Ni(NO_3)_2 + 2H_2O \tag{3}$$

$$2La_2O_3 + 8HNO_3 = 4La(NO_3)_2 + 4H_2O + O_2 \uparrow \tag{4}$$

2.  Carboxylation of MWCNTs. The purified MWCNTs were treated with 60 mL mixture of $H_2SO_4$ and $HNO_3$ (3/1, *v/v*) under ultrasonication at 30 °C for 3 h. After that, the mixture was added into a beaker with 200 mL ultrapure water and then cooled to room temperature. Then, the obtained suspension was centrifuged at 6000 rpm for 10 min, 8000 rpm for 5 min, and 10,000 rpm for 5 min successively. Finally, the precipitation was washed with deionized water (until the pH of the filtrate tested neutral) and dried under vacuum at 80 °C for 8 h to get MWCNTs-COOH.
3.  The dispersity of raw and functional MWCNTs in aqueous solution are shown in Figure 1b. MWCNTs-COOH presented a well dispersed suspension, and the homogeneous dispersion maintained stability for a long time (i.e., over one week). The raw MWCNTs were poorly dispersed, and the suspension stratified in a short time (less than 12 h). This might be explained by the fact that the MWCNTs were chemically shortened after purification and functionalization, and the modified hydrophilic group (-COOH) enhanced the solubility of the composite. Good distribution of MWCNTs-COOH in aqueous solution was beneficial for further modifications by improving the uniformity and stability of drop-casting on the surface of GCE.

#### 2.2.2. Fabrication of the Nonenzymatic Lactic Acid Sensor

The CoPPc/MWCNTs-COOH nanocomposites were synthesized by the procedure shown in Scheme 1. First, 150 mg of CoPc was dispersed in 15 mL of absolute ethanol, and 50 mg of MWCNTs-COOH was mixed in 50 mL of absolute ethanol to prepare a homogeneous suspension. The MWCNTs-COOH suspension was added dropwise to the CoPc suspension under magnetic stirring at 30 °C for 5 h. The suspension was then centrifuged at 8000 rpm for 15 min, and the resulting product was washed with deionized water several times to obtain the precipitate. Finally, CoPPc/MWCNTs-COOH nanocomposites were obtained after the precipitate was dried at 60 °C for 2 h.

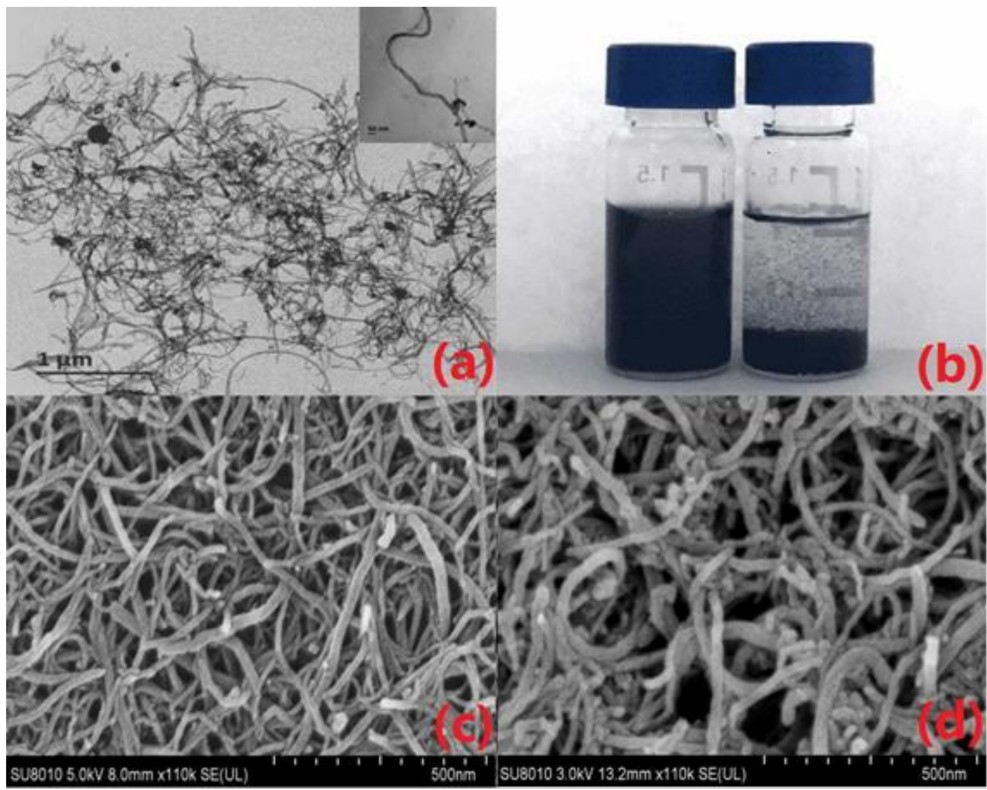

**Figure 1.** The morphology of the obtained CoPPc/MWCNTs-COOH: (**a**) TEM images of MWCNTs; (**b**) the suspension of raw and functioned MWCNTs; (**c**) SEM image of MWCNTs-COOH; (**d**) SEM image of CoPPc/MWCNTs-COOH.

Before modification, the GCE (diameter: 3 mm) was polished to mirror smoothness with 1.5 μm, 0.3 μm and 50 nm $Al_2O_3$ liquid paste. The polished electrode was successively ultrasonically cleaned in piranha solution (30% $H_2O_2$/98% $H_2SO_4$, 1/3, *v/v*), deionized water, and absolute ethanol (3 times, 0.5 min each time). It was then dried using high-purity nitrogen. First, 16 mg of CoPPc/MWCNTs-COOH was dispersed in 5 mL of DMF containing 5% Nafion and magnetically stirred at 60 °C for 1 h. Then, 2 μL of the suspension was dropped onto the surface of the GCE and dried under an infrared lamp (the dropping and drying procedures repeated five times). Finally, the CoPPc/MWCNTs-COOH/GCE was washed with deionized water and dried overnight.

### 2.3. Material Characterizations and Electrochemical Tests

The functional groups of the modified nanomaterials were identified using an AVA TAR370 Fourier transform infrared spectrometer (FTIR, Nicolet, Thermo Fisher Scientific, Waltham, MA, USA). The optical absorbance and band gap of the nanocomposite were measured using an UV-2550 UV-vis diffuse reflectance spectrophotometer (UV-vis-DRS, Shimadzu, Japan). The crystalline structures of nanocomposite were characterized using an Escalab 250Xi X-ray photoelectron spectroscopy (XPS, Thermofisher, Waltham, MA, USA). The morphology and surface element of CoPPc/MWCNTs nanocomposite were investigated by means of a SU-8010 Scanning Electron Microscopy (SEM, Hitachi, Tokyo, Japan). The impedance characteristics of bare electrode and CoPPc/WCNTs-COOH /GCE were evaluated using a PARSTAT 3000A electrochemical worked station (EWS, AMETEK, Berwyn, PA, USA). All the electrochemical experiments were performed on the EWS with a standard three-electrode configuration, which consisted of a CoPPc/H-MWCNTs-COOH/GCE working electrode, an Ag/AgCl reference electrode (3 M saturated KCl, diameter 2 mm) and a platinum auxiliary electrode (length 5 mm, diameter 1 mm). All the measurements were carried out in 0.1 M phosphate buffer saline at room temperature.

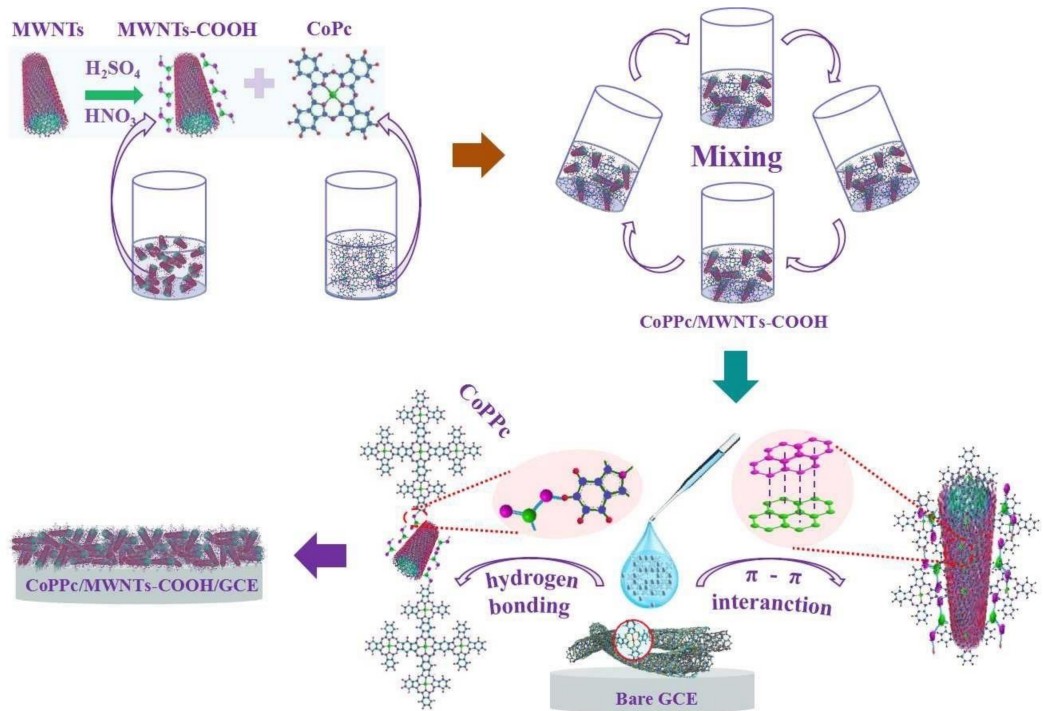

**Scheme 1.** The fabrication process of CoPPc/MWCNTs-COOH/GCE.

### 2.4. Detection of Lactic Acid in Rice Wine Samples

The sensor was applied to detect lactic acid in rice wine samples supplied by Shaoxing NuErHong Co., LTD (Shaoxing, China). In this experiment, 0.5 mL of each rice wine was mixed with 40 mL of 0.1 M PBS and the stable value of the amperometric responses was measured five times at a potential of 0.5 V. The lactic acid concentration of each rice wine sample was calculated based on the amperometric measurements and compared to the standard determination method of lactic acid.

### 3. Results

### 3.1. Characterization of CoPPc/MWCNTs-COOH Nanocomposites

The morphology of CoPPc/MWCNTs-COOH was characterized by SEM. As shown in Figure 1c,d, the overall morphology of the nanocomposite was uniform and presented the classical one-dimensional structure. The diameter of each nanocomposite fiber was thicker than that of functionalized MWCNT. The main contents of elemental C in the side walls of the MWCNTs comprised C=C bonds (sp$^2$), which presented a high degree of $\pi$ electron delocalization. Additionally, $\pi$-$\pi$ interactions played a key role in the association of aromatic compounds with natural organic substances. MWCNTs-COOH worked as a carrier in the synthesis process, and the CoPPc were covalently attached to the side wall of the MWCNTs through $\pi$-$\pi$ interactions (few connection with the H hydrogen bond).

Figure 2a presents FTIR spectroscopy of MWCNTs and MWCNTs-COOH, recorded between 400 and 4000 cm$^{-1}$ using KBr pellets. The peaks at 3418.33 cm$^{-1}$ corresponded to the stretching vibrations of -OH groups due to the H$_2$O attached to the surface of the MWCNTs. The bands appeared at 1638.16 cm$^{-1}$ and 1636.22 cm$^{-1}$ were assigned to C=C stretching vibrations of the aromatic rings. The peak values were attributed to the presence of five-membered rings at either end of concentric graphitic cylinders, while the seven-membered rings gave rise to bent nanotubes. A significant peak corresponding to C=O vibrations in carboxy groups was observed at 1711.23 cm$^{-1}$, which indicated that carboxyl groups were formed by the MWCNT functionalized process. The successful synthesis of CoPPc/MWCNTs-COOH nanocomposites was further verified using UV-vis absorption spectroscopy with DMF solvent (Figure 2b). No absorption peak was observed when the

test was performed on the raw MWCNTs, because there were no conjugate structures therein. CoPPc compounds typically exhibit two UV absorption peaks: the B-band and the Q-band. The B-band in the ultraviolet region presents strong absorption, and the Q-band in the visible region presents weak absorption. After the synthesis of CoPPc/MWCNTs-COOH nanocomposites, new absorption bands were observed at approximately 667 nm and 335 nm, corresponding to the characteristic Q-band and B-band of CoPPc. Therefore, the combination of CoPPc and MWCNTs-COOH was confirmed by UV-vis absorption spectroscopy. These observations were in good agreement with the XPS patterns. The chemical composition of the as-synthesized CoPPc/MWCNTs-COOH nanocomposites (the CoPPc/MWCNTs samples were studied in powdered and compacted forms) was further characterized by XPS. As shown in Figure 2c, the XPS spectrum clearly presented peaks corresponding to C, N, Co, O; the elemental C content was about 82.64%. The XPS survey spectra in the full range of 0–1200 eV demonstrated that there were no other metallic catalyst impurities except for elemental Co in the obtained CoPPc/MWCNTs-COOH, which confirmed the phase purity of the MWCNTs. It could be seen from C1s spectrum that CoPPc/MWCNTs-COOH was composed of three allotropes of carbon, and the three peaks at 284.79 eV, 287.21 eV, 255.62 eV for C1s corresponded to C=C (C-C), C=O, and C-O bonds (Figure 2d). In conclusion, the XPS patterns revealed that the MWCNTs had been efficient functionalized with carboxyl group (albeit containing fewer oxygen-containing functional groups).

### 3.2. Electrochemical Properties of the Modified CoPPc/MWCNTs-COOH/GCE

The electrochemical behavior of lactic acid was investigated by carrying out CV with the bare GCE and CoPPc/MWCNTs-COOH/CPE. Figure 3a presents cyclic voltammograms for the electrochemical reduction of 100 μM lactic acid with the MWCNTs/GCE and modified GCE in a PBS of pH 4.0, recorded at a fixed scan rate of 50 mV s$^{-1}$. No obvious redox peak of lactic acid appeared in the MWCNTs/GCE GCE (curve a), which indicated poor electron transfer at the interface of the bare electrode and inferior electrocatalytic performance of the bare GCE for the detection of lactic acid. The presentation of curve b demonstrated that the CoPPc/MWCNTs-COOH nanocomposite significantly accelerated electron transfer and improved peak shapes the best (Figure 4). This could be attributed mainly to the large surface area, the subtle electronic properties of MWCNTs and the ion exchange characteristics of CoPPc. Additionally, it could be seen that the anodic peak potentials of lactic acid appeared at a potential of about −0.18 V, and the obvious peak currents were clear evidence of the catalyzing effect of CoPPc/MWCNTs-COOH in the electroreduction of lactic acid.

Electrochemical impedance spectroscopy (EIS, frequency range: 0.1 Hz~10k Hz, amplitude: 0.01V RMS, and potential: 0.23 V) was used to study the features and impedance changes in the electronic conductivity of GCE after modification with CoPPc/MWCNTs-COOH. Figure 3b shows the Nyquist plots for the bare GCE and CoPPc/MWCNTs-COOH/GCE in the presence of 5 mM Fe(CN)$_6$$^{3-/4-}$ and 100 mM KCl. In the equivalent circuit, $R_s$, $R_{ct}$, $R_1$, and constant phase angle element (CPE1) represent the solution resistance, Faradaic charge-transfer impedance, diffusion impedance, and capacitance of the double layer, respectively. The $R_{ct}$ and CPE1 values, which correspond to the electron transfer resistance and double layer capacitance of the electrode surfaces, respectively, influenced the semicircle diameter and the slope of the Nyquist plot. As shown in Figure 3b, the bare GCE showed semicircular portions and linear parts in the high- and low-frequency regions, respectively, with the $R_{ct}$ value estimated to be 200 Ω (with the error of 0.58%). After the nanocomposite modification of the GCE, a much smaller semicircle (the $R_{ct}$ value decreased dramatically to 0.19 Ω (with the error of 6.43%)) and a sharp straight line appeared in the high- and low-frequency regions, respectively. This observation further confirmed that the electron transfer resistance decreased and the capacitance increased with the addition of the CoPPc/MWCNTs-COOH nanocomposite film. Therefore, a higher electroactive surface area of the 3D networked structure of nanocomposite film offered more active sites

for the catalytic reduction of lactic acid, and efficiently promoted electron transfer during electrochemical reaction.

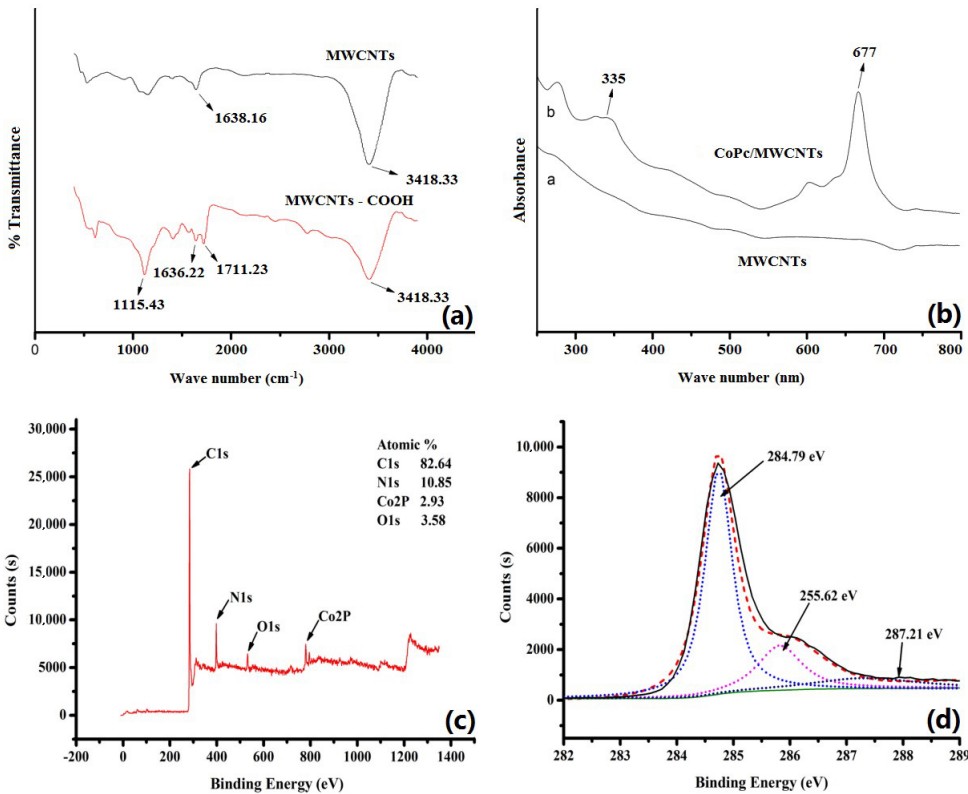

**Figure 2.** FTIR spectra of MWCNTs and MWCNTs-COOH (**a**), UV-vis absorption spectra of raw MWCNTs and CoPPc/MWCNTs-COOH nanocomposites (**b**); XPS spectra of CoPPc/MWCNTs-COOH (**c**), and XPS spectra of C1s (**d**).

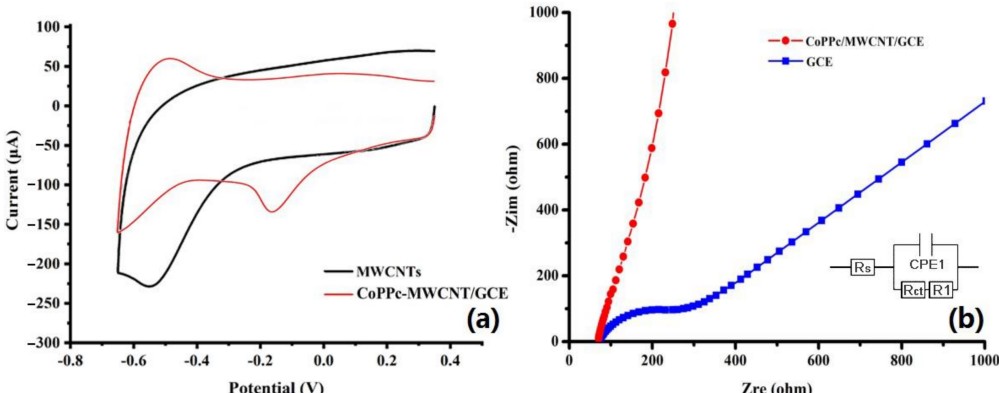

**Figure 3.** Cyclic voltammograms of 100 μM lactic acid at the MWCNTs/GCE and CoPPc/MWCNTs-COOH/GCE in PBS of pH 4.0, scan rate: 50 mV s$^{-1}$ (**a**); The EIS spectra at the bare GCE and CoPPc/MWCNTs-COOH/GCE in the presence of the mixture of 5 mM Fe(CN)$_6^{3-/4-}$ and 100.0 mM KCl (**b**).

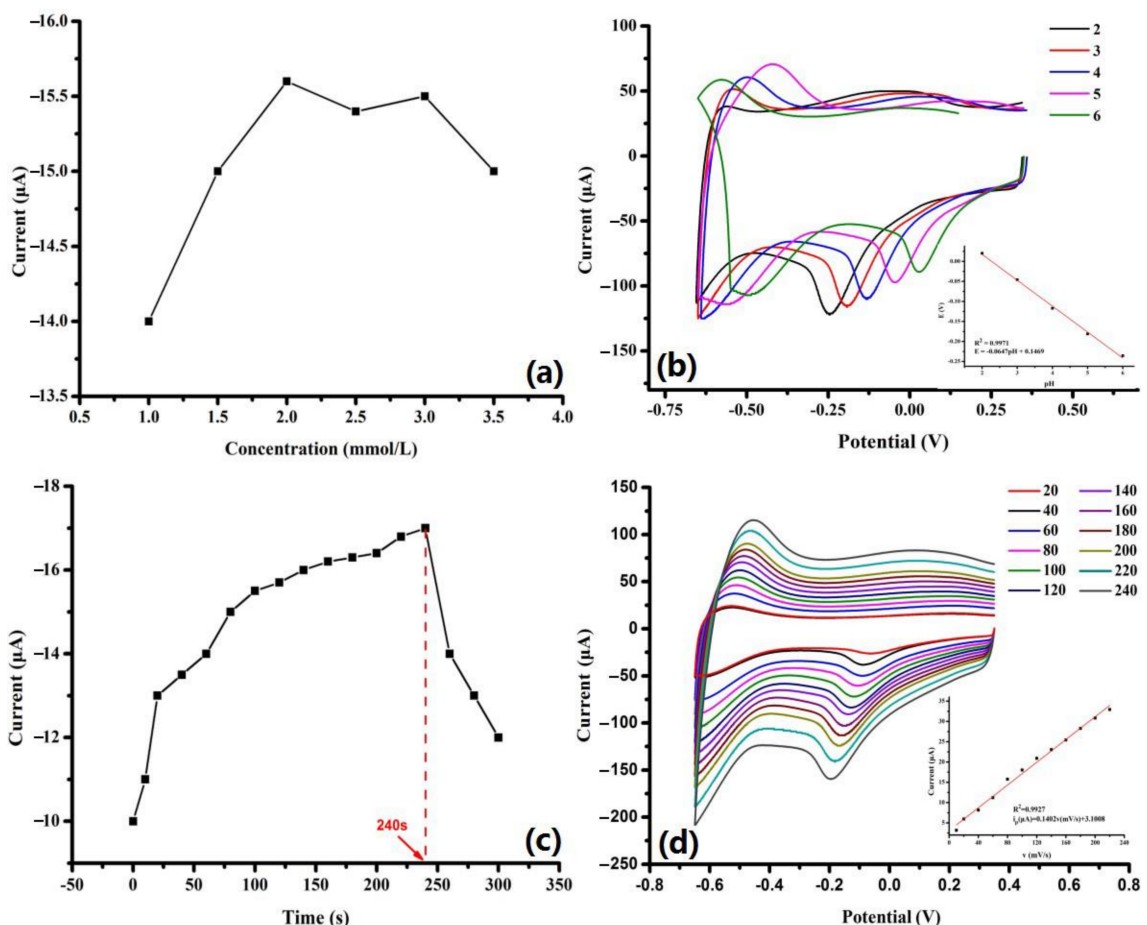

**Figure 4.** Effect of modifier concentration based on the CV in 100 μmol/L of lactic acid (pH 4.0) at 100 mVs$^{-1}$ from −0.65 to 0.35 V (**a**), pH value of electrolytic solution based on the electroreduction of lactic acid (0.1 mmol/L) tested by various pH values (ranging from 2 to 6) at 100 mV/s (**b**), accumulation time from 0 to 300 s on the electrochemical response of 50 μmol/L lactic acid (**c**) and the CV curves recorded in PBS (pH 4.0) containing 100 μmol/L lactic acid at various scan rates from 20 to 240 mV/s (**d**).

### *3.3. Optimization of Experimental Parameters*

#### 3.3.1. Influence of Modifier Dosage

It is well known that a higher modifier concentration can increase the powder porosity, resulting in a higher catalytic surface area and larger redox current. However, previous studies have demonstrated that increasing the modifier concentration beyond a certain point does not always yield the desired effect. Different concentrations of CoPPc/MWCNTs-COOH nanocomposites were dispersed in DMF (containing 0.05% Nafion) to prepare mixtures of 1, 1.5, 2, 2.5, 3, and 3.5 mg/mL. Each suspension (5 μL) was drop-cast onto the surface of the cleaned GCE, and the reduction state variation of the modified materials was probed with CV in 100 μmol/L of lactic acid (pH 4.0) at 100 mVs$^{-1}$ from −0.65 to 0.35 V. The relationship between the redox peek current and the modifier concentration is shown in Figure 4a. The redox peak current of lactic acid increased considerably at the beginning, correlating with the increasing concentration of CoPPc/MWCNTs-COOH from 1 to 2 mg/mL. Thereafter, the redox peak current decreased with further increases in modifier concentration. Thus, the optimal modifier concentration of CoPPc/MWCNTs-COOH appeared to be 2 mg/mL when the current peak was at maximum. A thicker modified film with abundant pores would provide larger surface areas and more electrochemical active sites, and more double-layer capacitors would form at the interface between modified film. However, electrochemical active sites would become saturated when the concentration

of CoPPc/MWCNTs-COOH was excessive, and the redox peak current would no longer increase with the further addition of modifier. Besides, over drop-casting could influence the stability and repeatability of the modified electrode because the modifier would be stripped away from the surface of the electrode during the electrochemical process, and the stripping process of the modifier might lead to a sharp decrease in the catalytic currents. According to the experimental analysis, 2 mg/mL was applied as the optimized concentration of CoPPc/MWCNTs-COOH to fabricate the modified electrode.

### 3.3.2. Influence of pH

The morphology of the synthesized metal oxides strongly depended on the quantity of $H^+$ or $OH^-$ ions present, which effectively determined the polymerization extent of the metal-oxygen bonds. The precursor solution pH affected the solubility of the precursor and the dissoluble/depositional ratio of configurational ions, which was highly correlated with the particle size and morphology of the nanocomposite structure. The current response of the electroreduction of lactic acid (0.1 mmol/L) at the CoPPc/MWCNTs-COOH/GCE was optimized by testing various pH values (ranging from 2 to 6) at 100 mV/s. As shown in Figure 4b, both the cathodic and anodic peak potentials shifted to more negative values and the peak currents decreased with increasing pH value, which indicated that protons were taking part in the electrocatalytic reduction process on the modified electrode. The reduction peak potential is linear with the pH value ($R^2$ = 0.9971). The linear equation is as follows:

$$E = -0.0647pH + 0.1469 \tag{5}$$

The slope of $-64.69$ mV/pH was close to the theoretical value of $-58.6$ mV/pH, which indicated that the Nernstian electrochemical reaction occurred with the involvement of an equal number of protons and electrons. The maximum reduction peak current for lactic acid was observed at pH 4. Thus, a pH of 4 was selected for the further studies.

### 3.3.3. Effect of Accumulation Time

In the accumulation step, the analytes were accumulated for hundreds of seconds on the electrode to increase both the mass of the material and the magnitude of the redox currents in the electrochemical reaction. In this study, the kinetics of lactic acid reduction at the CoPPc/MWCNTs-COOH/GCE was proven to be an adsorption-controlled process, and the accumulation step was applied to increase the detection sensitivity. The influence of the open-circuit accumulation time from 0 to 300 s on the electrochemical response of 50 μmol/L lactic acid was evaluated. The peak current accumulation time profile is shown in Figure 4c. It can be seen that the cathodic peak current increased dramatically with the increase in accumulation time from 0 to 100 s. The cathodic peak current increased slowly with increasing accumulation time from 100 to 240 s, which was probably due to the adsorptive equilibrium and accumulated saturation on the CoPPc/MWCNTs-COOH/GCE surface. The cathodic peak current decreased sharply with a further increase in the accumulation time from 240 to 300 s. The changes in the cathodic peak current might be attributed to the formation of adsorbed species on the electrode surface, which reduced the stability of the modified electrode after a long period of electrochemical accumulation. The three-dimensionally interpenetrating network of the CoPPc/MWCNTs-COOH nanocomposite provided a good framework for the diffusion of the target analyte with prolonged reaction times, and the nanocomposite network led to the strong adsorption of lactic acid on the electrode surface. From these results, 240 s was selected as the optimum accumulation time and was used in the subsequent experiments.

### 3.3.4. Effect of Scan Rate

In this study, different potential scan rates were used during CV to determine the electrochemical mechanism of lactic acid reduction. Figure 4d displayed the CV curves of CoPPc/MWCNTs-COOH/GC recorded in PBS (pH 4.0) containing 100 μmol/L lactic acid at various scan rates from 20 to 240 mV/s. It was obvious that the cathodic peak currents

increased with increasing potential scan rate, ranging from 20 to 240 mV/s. In addition, the cathodic peak current versus the scan rates has a good linear relationship ($R^2$ = 0.9927); the linear regression equation is as follows:

$$i_p = 0.1402v(\text{mVs}^{-1}) + 3.1008 \tag{6}$$

This result indicates that a typical adsorption-controlled electrochemical process occurred on the CoPPc/MWCNTs-COOH/GCE surface. The cathodic peak currents of the modified electrode still appeared in the blank buffer solution after the electrochemical reaction in the reactant solution, indicating that the modified electrode had a good adsorption capacity of lactic acid. Considering that a higher scan rate decreased the steady-state of the CoPPc/MWCNTs-COOH/GCE, 100 mV/s was selected as the optimized scan rate for further electrochemical experiments.

### 3.4. Kinetic Analysis of Lactic Acid on Modified Electrode
#### 3.4.1. The Number of Electrons and Protons Participated in the Electrocatalysis Process

As discussed in Section 3.3.2, the electrocatalysis process of lactic acid fitted (58.6 mV/pH at 25 °C) the Nernst behavior the best, and the electrochemical route included the same number of electrons and protons. As shown in Figure 4d, the irreversible cathodic peak shifted to a more positive position with increasing scan rates. The peak potential ($E_p$) and the logarithm of scan rate (ln $v$) with good linear relationship were obtained in the whole range studied (20–240 mV/s); the linear regression was calculated as follows:

$$E_p(V) = 0.1565 - 0.0386\log v \tag{7}$$

According to Laviron [41], for an irreversible electrochemical process, the relationship between peak potential ($E_p$) and the logarithmic scan rate (ln v) may be defined by $E_p$:

$$E_p = E^{0'} + \frac{2.303RT}{2\alpha n_\alpha F}\log v + \frac{2.303RT}{2\alpha n_\alpha F}\log\frac{\alpha n_\alpha F}{RTKs} \tag{8}$$

where $T$ is the temperature (in our case, room temperature, 298 K), $F$ is the Faraday constant (96,487 C/mol), $n_\alpha$ is the electron transfer number, and $2.303RT/2\alpha n_\alpha F$ is the slope of Tafel equation. The $\alpha n_\alpha$ value could be calculated using the slope of the curve ($2.303RT/2\alpha n_\alpha F$ = 0.0386), and $\alpha n_\alpha$ = 0.98. The electron transfer coefficient $\alpha$ was assumed to be 0.5 (the electron transfer coefficient for most electrochemical reactions on electrode surface was $0.25 < \alpha < 0.75$) for an irreversible electrode process. Then, the number of electrons transferred could be calculated as 2 ($n_\alpha$ = 2). Consequently, the calculation indicated that the redox reaction was a two-electron and two-proton transfer reaction. The possible electrocatalysis mechanism of lactic acid reduction on the surface of CoPPc/MWCNTs-COOH/GCE is illustrated in Scheme 2.

#### 3.4.2. The Effective Surface Area of the Modified Electrode

Chronocoulometry (CC) is an accurate electrochemical method for the detection of the analytes adsorbed on an electrode surface. The advantage of CC over other electrochemical methods is that the intercept at time zero obtained from extrapolation is the sum of the double-layer charge. The charge values are due to the reaction of the analyte adsorbed on the electrode surface, and can be clearly separated from the data related to the quantity of analyte in solution. In this study, the electrochemically effective surface areas ($A$) of GCE and CoPPc/MWCNTs-COOH/GCE were investigated by chronocoulometry using $1.0 \times 10^{-3}$ mol/L $K_3[Fe(CN)_6]$ as a model complex (the diffusion coefficient of $K_3[Fe(CN)_6]$ in 1 M KCl. The linear dependence between current values and $t^{1/2}$ can be described as follows:

$$Q(c) = -2.13 \times 10^{-6} + 1.47 \times 10^{-4}t^{\frac{1}{2}} \tag{9}$$

The current corresponding to the electrochemical reaction also can be described by the Cottrell equation:

$$Q(c) = 2nFAC_0 D_0^{\frac{1}{2}} t^{\frac{1}{2}} \pi^{-\frac{1}{2}} \tag{10}$$

where $A$ is the surface area of the working electrode, $n$ is the number of transferred electron ($n$ = 2 for lactic acid), $F$ is the Faraday constant (96,487 C/mol), $C_0$ is the bulk concentration of $K_3[Fe(CN)_6]$ solution, and $D_0$ is the diffusion coefficient of $K_3[Fe(CN)_6]$ solution ($D_0$ = 7.6 × 10$^{-6}$ cm$^2$/s). Based on the slope of the linear relationship between $Q$ and $t^{1/2}$, $A$ can be calculated when $C$, $D$ and $n$ are known. Therefore, chronocoulometry was performed based on the above experimental parameters. According to Cottrell's equation and the slope values, $A$ was calculated to be 0.037 cm$^2$ and 0.49 cm$^2$ for bare GCE and CoPPc/MWCNTs-COOH/GCE, respectively. These results indicate that the effective surface area of electrode increased after electrode modification, and the increased surface also enhanced the current response, improved the detection sensitivity, and decreased the limit of detection of the modified electrode.

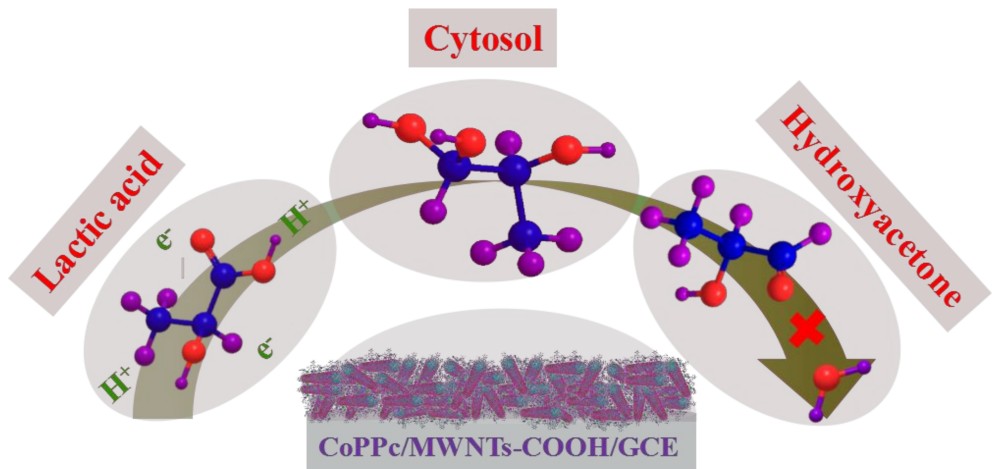

**Scheme 2.** Electrocatalysis mechanism of lactic acid reduction on the surface of modified electrode.

3.4.3. The Analysis of Diffusion Coefficient

The CC responses of CoPPc/MWCNTs-COOH/GCE from solutions prepared in the presence of 1.0 × 10$^{-3}$ mol/L lactic acid (dissolved in 0.1 mol/L PBS with pH = 4) were determined, and the linear dependences were obtained by plotting net current values versus $t_{1/2}$ as:

$$Q(c) = -3.03 \times 10^{-6} + 5.12 \times 10^{-4} t^{\frac{1}{2}} \tag{11}$$

Thus, according to Cottrell's Equations (3)–(6) and the slopes of the $Q$ vs $t^{1/2}$ curves ($A$ = 0.49 cm$^2$), the diffusion coefficient ($D$) of lactic acid could be calculated to be 9.22 × 10$^{-5}$ cm$^2$/s.

*3.5. Determination of Lactic Acid Concentration*

Based on the optimum electrochemical parameters, linear sweep voltammetry (LSV) was used to evaluate the efficiency of the CoPPc/MWCNTs-COOH/GCE sensor for the determination of different concentrations of lactic acid in 0.1 mol/L phosphate buffer saline (pH 4.0). As shown in Figure 5, the cathodic peak current increased linearly as the lactic acid concentration increased from 10 to 240 µmol/L. The linear relationship between cathodic peak current and concentration of lactic acid could be calculated according to the following regression equation:

$$i_p(\mu A) = 0.1379c(\mu mol/L) + 0.0572 \tag{12}$$

where the correlation coefficient could be calculated as 0.9988, and the LOD of 1 × 10$^{-6}$ mol/L could be estimated when the SNR was 3.

The comprehensive performance of CoPPc/MWCNTs-COOH/GCE was better than that of most reported sensors for the detection of lactic acid, as shown in Table 1. The excellent performance can be attributed to the combined effects of the high conductivity of the MCNTs-COOH and the electro-catalytic activity of CoPPc.

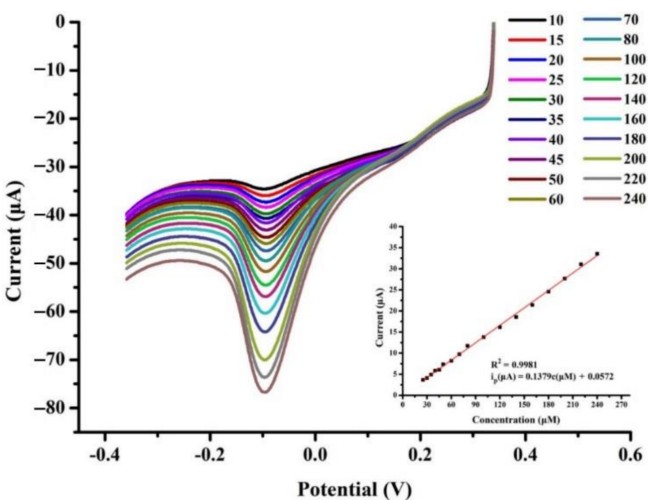

**Figure 5.** The CoPPc/MWCNTs-COOH/GCE sensor for the determination of different concentrations of lactic acid in 0.1 mol/L PBS (pH 4.0).

**Table 1.** List of recently reported lactic acid sensors with the sensor fabricated in the present study.

| Electrode | Sensor Type | Sensitivity ($\mu A/mM \cdot cm^2$) | LOD ($\mu M$) | Linear Range (mM) | Ref. |
|---|---|---|---|---|---|
| MoS$_2$-AuPt@SPE | Nonenzymatic | - | 0.33 | 0.005–3 | [42] |
| FePt NPs-g-C$_3$N$_4$/CZO | Nonenzymatic | - | 110 | 0.2–10 | [43] |
| Co-NCF composite | Nonenzymatic | 1068 | 13.7 | 0.1–1 | [44] |
| AuNPs-ERGO-PAH | Enzymatic | 0.28 | 1 | 0.5–3 | [45] |
| GC/NiO@Au | Nonenzymatic | 8 | 11.6 | 0.1–500 | [46] |
| NAD$^+$ | LDH | PEDOT-TiONWs | FTO | Enzymatic | 0.1386 | 0.08 | 0.0005–0.3 | [47] |
| AuNP-cysteamine-LDH | Enzymatic | 73.16 | 411 | 0.5–7 | [48] |
| ZIF-67 derived NiCo LDH | Nonenzymatic | 83.98 | 399 | 2.0–26.1 | [49] |
| CoPPc/MWCNTs-COOH/GCE | Nonenzymatic | - | 2 | 0.01–0.24 | This work |

### 3.6. Reproducibility, Stability, and Interference Analysis

The reproducibility of the CoPPc/MWCNTs-COOH/GCE was analyzed in the presence of 1.5 mM lactic acid in 0.1 M PBS (pH 4). Differential pulse voltammetry experiments were repeated five times with the same lactic acid sensor. The relative standard deviation (RSD) was 2.8%, indicating that the reproducibility of the lactic acid sensor was excellent. Similarly, the fabrication reproducibility was investigated using ten different lactic acid sensors prepared under the same conditions. Each CoPPc/MWCNTs-COOH/GCE was detected in the same electrochemical cell, and all the sensors exhibited similar response currents with a relative standard deviation of no more than 4.6%. Therefore, the optimized fabrication process was highly reliable with reproducible results. The stability of the sensor was also evaluated by intermittent measurement of lactic acid (1.5 mM) once a day. As shown in Figure 6a, the current values did not change in the first four days, and the sensors maintained 93.6% of their initial current response after 6 days at room temperature. The response current decreased to 83.4% of its initial response after 14 days. However, each CoPPc/MWCNTs-COOH/GCE was able to be used immediately after fabrication, and the stability of the lactic acid sensor in the first week was sufficient for detection.

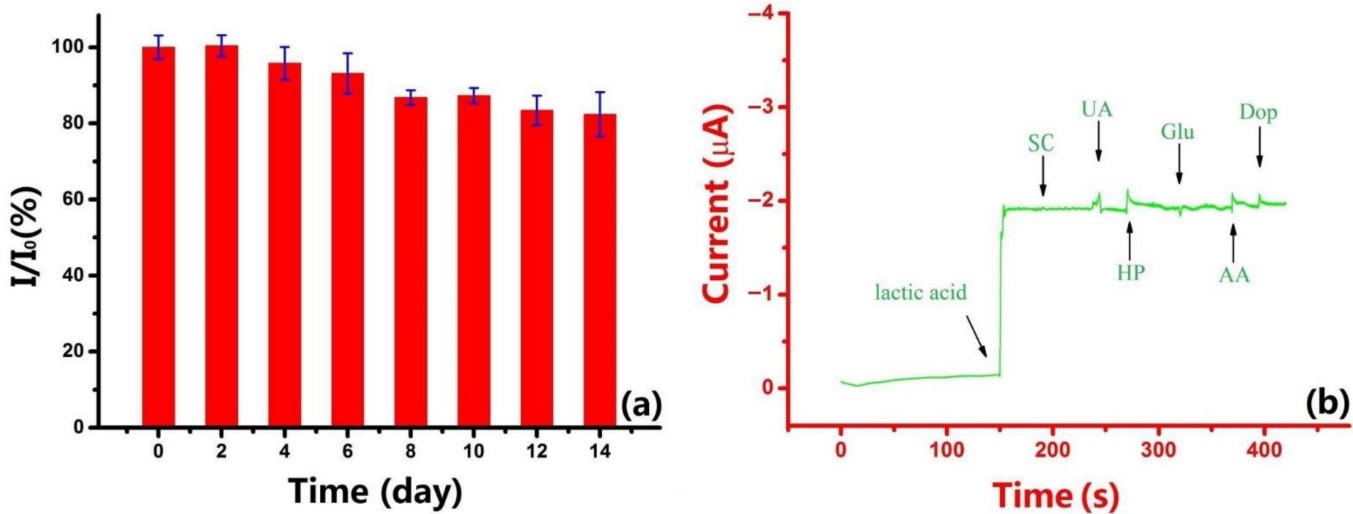

**Figure 6.** Stability test (**a**) and interference test (**b**) of the fabricated lactic acid sensor in a 100-fold concentration of Na$^+$ and Cl$^-$, 10-fold concentration of SC, AA, UA and Dop, and in the same concentration of Glu and HP.

The elimination of possible interfering electrochemical responses of easily oxidizable species (i.e., coexistence substances) in rice wine is highly significant for the practical application of lactic acid sensors (Figure 6b). In this study, the effectiveness of CoPPc/MWCNTs-COOH/GCE was evaluated using amperometric i-t curves for the simultaneous determination of ascorbic acid (AA), uric acid (UA), dopamine (Dop), sodium chloride (SC), glucose (Glu) and hydrogen peroxide (HP) under a potential of −0.23V with stirring in phosphate buffer saline (pH 7.0). The results showed that the following did not interfere with the detection of the lactic acid peak current changes: Na$^+$ and Cl$^-$ at a 100-fold concentration; AA, UA, and Dop at a 10-fold concentration; and Glu and HP at the same concentration as lactic acid. Although the cathode currents of UA, HP, and AA were slightly larger, all the redox peak currents were less than ±5%. The interference analysis suggested that the fabricated sensor had good anti-interference ability.

### 3.7. Real Samples Analysis

In order to illustrate the applicability of the fabricated CoPPc/MWCNTs-COOH/GCE, linear sweep voltammetry (LSV) with the optimal conditions was applied to detect lactic acid in rice wine samples (the rice wine samples were diluted in 0.1 M phosphate buffer saline). The results of each experiment are shown in Table 2. It can be seen that the lactic acid concentrations tested by the as-prepared sensors were in good agreement with those obtained by rice wine company based on the standard method [50], the recoveries ranged from 99.6% to 103.6% and the RSD of three repeated tests was less than 5.88%. These results indicate that the fabricated CoPPc/MWCNTs-COOH/GCE efficiently detected the lactic acid concentrations of standard lactic acid samples, and could be used to test the lactic acid concentrations of real rice wine samples with excellent veracity and reliability.

**Table 2.** Determination of lactic acid in rice wine samples.

| Rice Wine Samples | Lactic Acid Concentration (mM) | | Error (%) | Recovery (%) |
|---|---|---|---|---|
| | Reference Method | Sensor | | |
| 1 | 0.89 | 0.93 | 4.49% | 101.9% |
| 2 | 1.67 | 1.60 | −4.19% | 100.1% |
| 3 | 1.24 | 1.31 | 5.65% | 99.6% |
| 4 | 0.68 | 0.72 | 5.88% | 103.6% |

## 4. Discussion

In this work, a novel lactic acid sensor made of CoPPc/MWCNTs-COOH nanocomposite has been successfully fabricated. The synthesized nanocomposite presented a 3D structure and enlarged the active surface area of the electrode ($A$ = 0.49 cm$^2$), which provided more exposed electro-active sites and significantly improved the electrode diffusion coefficient ($D$ = 9.22 $\times$ 10$^{-5}$ cm$^2$/s). The nonenzymatic sensor exhibited excellent electro-catalytic activity toward lactic acid reduction and provided a linear range of 10–240 μmol/L with a limit of detection of 1 $\times$ 10$^{-6}$ mol/L. The proposed lactic acid sensing strategy was also successfully applied to determine the lactic acid content in rice wine, and revealed the advantages of high selectivity, good repeatability, and stability. Therefore, the CoPPc/MWCNTs-COOH/GCE sensor has great potential for lactic acid sensing in real samples.

**Author Contributions:** Z.W.: conceptualization, methodology, software. W.S.: data curation, writing-original draft preparation. J.M.: visualization, investigation. supervision. J.M.: writing-reviewing and editing. All authors have read and agreed to the published version of the manuscript.

**Funding:** This research was funded by the Chinese National Foundation of Nature and Science through Project 31972200.

**Data Availability Statement:** Not applicable.

**Acknowledgments:** The authors acknowledge the financial support of the Chinese National Foundation of Nature and Science through Project 31972200.

**Conflicts of Interest:** All other authors declare no competing financial interests.

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
