# Peer review of "Nonenzymatic Lactic Acid Detection Using Cobalt Polyphthalocyanine/Carboxylated Multiwalled Carbon Nanotube Nanocomposites Modified Sensor"

_chemosensors, doi:10.3390/chemosensors10020083_

Round 1

Reviewer 1 Report

In this work, the authors presented a non-enzymatic electrochemical sensor based on CoPPc/MWCNT-COOH for sensing lactic acid. In general, the sensor showed a wide detection range and a low detection limit (1x10-6 M/L); the manuscript is well presented and the results are interesting. In my opinion, the manuscript should be accepted for publication after some revisions:

  1. In the abstract, the authors claimed that the sample was characterized by XRD and the manuscript shows no results of this technique.
  2. Revise the English in the entire manuscript, there are misspelled words, grammar mistakes, missing symbols, and subscripts in various sections.
  3. Equations 1-4 need to be balanced and revised
  4. Figure 2(d) is the XPS of carbon, not XRD.
  5. From lines 227 to 241 authors mention XRD patterns but in the manuscript there is no XRD results. Include them.
  6. Include the equivalent circuits in Figure 3(b) 
  7. Caption in Figure 6 needs to be corrected
  8. Figure 6(b) requires additional comments.
  9. If possible, describe which techniques are used in the Hospital to quantify the lactic acid concentrations.
  10. Include a table comparing the performance of the sensor among other sensors and/or techniques.

Author Response

Comments and Suggestions for Authors 1

In this work, the authors presented a non-enzymatic electrochemical sensor based on CoPPc/MWCNT-COOH for sensing lactic acid. In general, the sensor showed a wide detection range and a low detection limit (1x10-6 M/L); the manuscript is well presented and the results are interesting. In my opinion, the manuscript should be accepted for publication after some revisions:

1) In the abstract, the authors claimed that the sample was characterized by XRD and the manuscript shows no results of this technique.

Thank you for the comment. The description is mistake, and the XRD was not used in the experiment. The content about XRD was deleted in the text, and the details was as follows (Page 1, line 13 and 14):

…Fourier transform infrared spectroscopy, ultraviolet-visible absorption spectroscopy, and X-ray photoelectron spectroscopy.

2) Revise the English in the entire manuscript, there are misspelled words, grammar mistakes, missing symbols, and subscripts in various sections.

Thank you for the comment. The English of the manuscript was improved by a native speaker, and all the revised parts were marked by red color in the text.

3) Equations 1-4 need to be balanced and revised

Thank you for the comment. The Equations were balanced as follows (Page 3, line 121):

3C + 4HNO3 = 3CO2 ↑ + 4NO + 2H2O

(1)

Ni + 4HNO3 = Ni(NO3)2 + 2NO2 ↑ + 2H2O

(2)

Ni + 4HNO3 = 2Ni(NO3)2 + 2H2O

(3)

2La2O3 + 8HNO3 = 4La(NO3)2 + 4H2O + O2

(4)

4) Figure 2(d) is the XPS of carbon, not XRD.

Thank you for the comment. XRD was replaced with XPS, and the detail was as follows (Page 6, line 197):

…and XPS spectra of C1s (d).

5) From lines 227 to 241 authors mention XRD patterns but in the manuscript there is no XRD results. Include them.

Thank you for the comment. XRD was replaced with XPS, and the details was as follows (Page 6, line 216-217; Page 7, line 227):

Page 6, line 216-217: These observations were in good agreement with the XPS patterns.

Page 7, line 227: In conclusion, the XPS patterns revealed that …

6) Include the equivalent circuits in Figure 3(b) 

Thank you for the comment. The equivalent circuit was included as follow (Page 7, line 245):

7) Caption in Figure 6 needs to be corrected

Thank you for the comment. Caption in Figure 6 was revised, and the details were as follows (Page 14, line 458-459):

Figure 6. Stability test (a) and interference test (b) of the fabricated lactic acid sensor in a 100-fold concentration of Na+ and Cl-, 10-fold concentration of SC, AA, UA and Dop, and in the same concentration of Glu and HP.

8) Figure 6(b) requires additional comments.

Thank you for the comment. The content about Figure 6(b) was revised, and the details were as follows (Page 13, Line 445-456)

The elimination of possible interfering electrochemical responses of easily oxidizable species (the co-existence substances) in rice wine is highly significant for the practical application of lactic acid sensors (Figure 6b). In this study, the effectiveness of CoPPc/MWCNTs-COOH/GCE was evaluated using amperometric i-t curves for the simultaneous determination of ascorbic acid (AA), uric acid (UA), dopamine (Dop), Sodium chloride (SC), Glucose (Glu) and hydrogen peroxide (HP) in phosphate buffer saline (pH 7.0). The results showed that the following did not interfere with the detection of the lactic acid peak current changes: Na+ and Cl- at a 100-fold concentration; AA, UA, and Dop at a 10-fold concentration; and Glu and HP at the same concentration as lactic acid. Although the cathode currents of UA, HP, and AA were slightly larger, all the redox peak currents were less than ± 5%. The interference analysis suggested that the fabricated sensor had a good anti-interference ability.

9) If possible, describe which techniques are used in the Hospital to quantify the lactic acid concentrations.

Thank you for the comment. the description about the Hospital and blood is not improper in the text, therefor, those content was already deleted. The techniques used for monitoring lactic acid level was the national standard method, and the details were as follows (Page 14, line 464-466):

It could be seen that the lactic acid concentrations tested by the as-prepared sensors were in good agreement with the values obtained by rice wine company based on the standard method [50], …

10) Include a table comparing the performance of the sensor among other sensors and/or techniques.

Thank you for the comment. Table 1 comparing the performance of the sensor among other sensors was already added into the text, and the details were as follows (Page 13, line 442-444):

Table 1. List of recently reported lactic acid sensors with the sensor fabricated in the present study.

Electrode

Sensor type

Sensitivity

(µA/mM·cm2)

LOD

(μM)

Linear range

(mM)

Ref.

MoS2-AuPt@SPE

Non-enzymatic

-

0.33

0.005-3

42

FePt NPs-g-C3N4/CZO

Non-enzymatic

-

110

0.2-10

43

Co-NCF composite

Non-enzymatic

1068

13.7

0.1-1

44

AuNPs-ERGO-PAH

Enzymatic

0.28

1

0.5-3

45

GC/NiO@Au

Non-enzymatic

8

11.6

0.1-500

46

NAD|LDH|PEDOT-TiONWs|FTO

Enzymatic

0.1386

0.08

0.0005-0.3

47

AuNP-cysteamine-LDH

Enzymatic

73.16

411

0.5-7

48

ZIF-67 derived NiCo LDH

Non-enzymatic

83.98

399

2.0-26.1

49

CoPPc/MWCNTs-COOH/GCE

Non-enzymatic

-

2

0.01-0.24

This work

Reviewer 2 Report

The authors describe preparation a non-enzymatic lactic acid sensor using cobalt phthalocyanine/carboxylated MWCNT composite material. Non-enzymatic sensor can be useful for food industry for online monitoring lactic acid production. Enzymatic sensors can suffer from low operational stability and temperature inactivation. Non-enzymatic sensors may offer a more practical and cost-efficient alternative. However, selectivity and sensitivity should be carefully evaluated in the model conditions and real matrices. The presented manuscript should be carefully revised to avoid misunderstanding and wrong data interpretation. Unfortunately, the current version of the manuscript is not well prepared for complete evaluation of its scientific quality. My recommendations and questions are listed below:

1) Unsubstituted metallophthalocyanines have very limited solubility in most of solvents. Does 10 mg/ml CoPc produce suspension in ethanol?

2) Scheme 1 and experimental part.

Oligomerization of the monomeric CoPc into the 5-meric fusion CoPc as it is depicted in Scheme 1 is unrealistic in the given experimental conditions. What could be the structural composition of the composite material?

3) Authors confuse electrochemical reduction (as said in abstract in some parts of the text, eg line 327) and oxidation of lactic acid (as said in some other parts of the text, eg line 245) at the modified electrode. Is the electrochemical data given in agreement with the IUPAC conventions? The figure and the discussion should be given in agreement with IUPAC definitions for the electrochemical processes.

4) Figure 3.

The authors should give CV in the same conditions for the modified electrode in the absence of lactic acid.

How does the CV look for  MWCNT/GCE in the absence and presence of 100 uM lactic acid?

5) Parameters for recording EIS are missing (frequency range, amplitude, potential).

What is the error for Rct values obtained from fits and given on page 8?

6) Figure 4.

The caption must provide all essential experimental conditions.

Is the axe name for Panel (a) correct?

7) Minor grammar corrections

Line 286: “It’s” should be “It is”

Line 295: extra space µ and mol and missing space in “0.35V”

Line 301: “peek” should be “peak”

8)  “Log”  should be “ Ln”  if the logarithm is natural.

9) Figure 5 and the text in the section 3.6.

The plot goes only till 240 µmol/L. The dependence of the sensor’s response as function of lactic acid concentration at higher concentrations was not tested. In the section 3.6 the authors say that the reproducibility and stability of the sensor was tested with “1.5 mM lactic acid”. The sensor could suffer from decline of the sensitivity (fouling) in the low concentration range while having less fluctuation in the very high saturated concentration.        

10) Figure 6

The caption misses description for panel (a).

11) Page 13, lines 463-464.

The authors say in the text that the selectivity tests was conducted using DPV but the corresponding figure shows continuous measurements in time, which cannot be the data from DPV according to the given time scale. This reveals clear disagreement between the text and figures.

The selectivity test must be conducted in the same conditions as the calibration curve and the real sample analysis.

12) Table 1. Determination of lactic acid in rice wine samples.

“Hospital” should be replaced by “reference method”

Information about this reference method must be also provided.

13) The samples were diluted by a factor of 81 (0.5 ml in 40 ml). The concentration is found in the range 0.7-1.6 mmol/L, meaning that the concentration after the dilution was 10-20 µmol/L. How the sensor performs at these concentrations of lactic acid? Figure 5 shows the graph starting from 24 µmol/L.

 14) Was the method in Section 3.7 the same as in Section 3.5 and the figure with the calibration curve (Figure 5).

Author Response

Comments and Suggestions for Authors 2

The authors describe preparation a non-enzymatic lactic acid sensor using cobalt phthalocyanine/carboxylated MWCNT composite material. Non-enzymatic sensor can be useful for food industry for online monitoring lactic acid production. Enzymatic sensors can suffer from low operational stability and temperature inactivation. Non-enzymatic sensors may offer a more practical and cost-efficient alternative. However, selectivity and sensitivity should be carefully evaluated in the model conditions and real matrices. The presented manuscript should be carefully revised to avoid misunderstanding and wrong data interpretation. Unfortunately, the current version of the manuscript is not well prepared for complete evaluation of its scientific quality. My recommendations and questions are listed below:

1) Unsubstituted metallophthalocyanines have very limited solubility in most of solvents. Does 10 mg/ml CoPc produce suspension in ethanol?

Thank you for the comment. The suspension was produced in the method, not solution. The details were corrected as follows (Page 4, line 142-143):

The MWCNTs-COOH suspension was added dropwise to the CoPc suspension under magnetic stirring at 30 °C for 5 h.

2) Scheme 1 and experimental part.

Oligomerization of the monomeric CoPc into the 5-meric fusion CoPc as it is depicted in Scheme 1 is unrealistic in the given experimental conditions. What could be the structural composition of the composite material?

Thank you for the comment. The fabrication process of the CoPPc/MWCNTs-COOH nanocomposites modified electrode was referred to several former work [1,2,3,4], and the detailed fabrication process was as follows:

150 mg of CoPc was added in 15 mL of absolute ethanol and dispersed by ultrasound. 50 mg of MWCNTs-COOH was mixed in 50 mL of absolute ethanol to prepare a homogeneous suspension. The MWCNTs-COOH suspension was added dropwise to the CoPc suspension under magnetic stirring at 30 °C for 5 h. The suspension was then centrifuged at 8000 rpm for 15 min, and the resulting product was washed with deionized water several times to obtain the precipitate. Finally, CoPPc/MWCNTs-COOH nanocomposites were obtained after the precipitate was dried at 60 °C for 2 h.

16 mg of CoPPc/MWCNTs-COOH was dispersed in 2 mL of DMF containing 5% Nafion for ultrasonic dispersion. Use a pipette gun to drop 2 μL of modifier on the surface of the glassy carbon electrode, bake it at 25-30cm under an infrared lamp for 40 min at a temperature not exceeding 45℃, and repeat 5 times. The modified electrode was dried for 1 h at room temperature, and the surface of the electrode was washed with deionized water to obtain CoPPc/MWCNTs/GCE, which was dried for later use.

References

  1. Li, Z.P.,P, Y.X.,Yang, S.F.,Zhang, R.,Li, K.,Zuo, X. Preparation and Oxygen Reduction Catalytic Performance of Iron-phthalocyanine Polymer /Multi-walled Carbon Nanotubes Composites. Chem J Chinese U, 2015, 36, 2016-2023.
  2. Wang, Y. The synthesis of carbon nanotubes-phthalocyanine dyes and investigation of catalytic performance. Changzhou University, 2016.
  3. P., Qin. C.X., Chen. L.M., Xia. X.Q. Study of carbon paste electrode modified with cobalt phthalocyanine for determination of bisphenol A. J Hubei Normal U, 2016, 36, 94-98.
  4. Balan, I., David, I.G., David, V., Stoica, A.I., Mihailciuc, C., Stamatin, I., Ciucu, A.A. Electrocatalytic voltammetric determination of guanine at a cobalt phthalocyanine modified carbon nanotubes paste electrode. J Electroanal Chem, 2011, 654, 8-12.

3) Authors confuse electrochemical reduction (as said in abstract in some parts of the text, eg line 327) and oxidation of lactic acid (as said in some other parts of the text, eg line 245) at the modified electrode. Is the electrochemical data given in agreement with the IUPAC conventions? The figure and the discussion should be given in agreement with IUPAC definitions for the electrochemical processes.

Thank you for the comment. Lactic acid was reduced on the modified electrode, so the “oxidation” was replaced with “reduction” in this manuscript. The details were as follows:

Page 1, Line 20-21: …towards lactic acid reduction over a wide concentration range (10 to 240 μM) …

Page 7, Line 233-234: Figure 3a depicts cyclic voltammograms for the electrochemical reduction of 100 μM lactic acid…

Page 7, Line 244: …toward the electroreduction of lactic acid.

Page 8, Line 267-268: …for the catalytic reduction of lactic acid and efficiently promoted electron transfer during electrochemical reaction.

Page 8, Line 282-283: …and the reduction state variation of the modified materials…

Page 9, Line 307-308: The current response of the electroreduction of lactic acid…

Page 9, Line 311: …that protons were taking part in the electrocatalytic reduction process…

Page 9, Line 312: The reduction peak potential is linear with the pH value…

Page 9, Line 316-317: The maximum reduction peak current for lactic acid was observed at pH 4.

Page 9, Line 321-323: In this study, the kinetics of lactic acid reduction at the CoPPc/MWCNTs-COOH/GCE…

Page 10, Line 340-341: In this study, different potential scan rates were used during CV to determine the electrochemical mechanism of lactic acid reduction.

Page 10, Line 371-373: The possible electrocatalysis mechanism of lactic acid reduction on the surface of CoPPc/MWCNTs-COOH/GCE was illustrated in Scheme 2.

Page 14, Line 478-480: The non-enzymatic sensor exhibited excellent electrocatalytic activity toward lactic acid reduction and provided a linear range of 10-240 μmol/L with a limit of detection of 1 × 10-6 mol/L.

The electrochemical data are in agree with the IUPAC conventions. The currents measured by the CHI660E were in opposite directions, so the data were processed.

4) Figure 3. The authors should give CV in the same conditions for the modified electrode in the absence of lactic acid. How does the CV look for MWCNT/GCE in the absence and presence of 100 uM lactic acid?

Thank you for the comment. The CV curves of MWCNT/GCE in the absence and presence of 100 uM lactic acid are as follows:

It was observed that the response current of MWCNT/GCE wasweak, and the MWCNT materials had no catalytic effect on lactic acid. Therefore, the CV curves of MWCNT/GCE in the presence of 100 uM lactic acid was not presented in the manuscript.

5) Parameters for recording EIS are missing (frequency range, amplitude, potential). What is the error for Rct values obtained from fits and given on page 8?

Thank you for the comment. The missing parameters were already added into the text, and the details were as follows (Page 7, line 249 - 252; Page 8, line 260 - 262):

Page 7, line 249 - 252: Electrochemical impedance spectroscopy (EIS, frequency range: 0.1Hz~10k Hz, amplitude: 0.01V RMS, and potential: 0.23 V) was used to study the features and impedance changes of electronic conductivity after the modification of CoPPc/MWCNTs-COOH.

Page 8, line 260 - 262: with the Rct value estimated to be 200 Ω (with the error of 0.58%). After the nanocomposite modification of the GCE, a much smaller semicircle (the Rct value decreased dramatically to 0.19 Ω (with the error of 6.43%)) and a sharp straight line were displayed in the high- and low-frequency regions, respectively.

6) Figure 4. The caption must provide all essential experimental conditions.

Is the axe name for Panel (a) correct?

Thank you for the comment. The essential experimental conditions was provided in the caption of Figure 4, and the incorrected caption was already revised. The details were as follows (Page 8, line 270-273):

Figure 4. Effect of modifier concentration based on the CV in 100 μmol/L of lactic acid (pH 4.0) at 100 mVs-1 from -0.65 to 0.35 V (a), pH value of electrolytic solution based on the electroreduction of lactic acid (0.1 mmol/L) tested by various pH values (ranging from 2 to 6) at 100 mV/s (b), accumulation time from 0 to 300 s on the electrochemical response of 50 μmol/L lactic acid (c) and the CV curves recorded in PBS (pH 4.0) containing 100 μmol/L lactic acid at various scan rates of 20-240 mV/s (d).

7) Minor grammar corrections

Line 286: “It’s” should be “It is”

Line 295: extra space µ and mol and missing space in “0.35V”

Line 301: “peek” should be “peak”

Thank you for the comment. The grammar was revised as follows:

Page 8, Line 276: It is well known that a higher modifier concentration can increase…

Page 8, Line 283 - 284: …in 100 μmol/L of lactic acid (pH 4.0) at 100 mVs-1 from -0.65 to 0.35 V…

Page 9, Line 289-290: …when the current peak achieved the maximum.

The English of the manuscript was improved by a native speaker, and all the revised parts were marked by red color in the text.

8)  “Log”  should be “ Ln”  if the logarithm is natural.

Thank you for the comment. Log was replaced with Ln. The details were as follows (Page 10, line 360, line 363):

Page 10, line 360: …the logarithm of scan rate (ln v) with good linear relationship…

Page 10, line 363: …and the logarithmic scan rate (ln v) could be defined…

9) Figure 6

The caption misses description for panel (a).

Thank you for the comment. The description for panel (a) was added, and the details were as follows (Page 14, lines 458):

Figure 6. Stability test (a)…

10) Page 13, lines 463-464.

The authors say in the text that the selectivity tests was conducted using DPV but the corresponding figure shows continuous measurements in time, which cannot be the data from DPV according to the given time scale. This reveals clear disagreement between the text and figures.

The selectivity test must be conducted in the same conditions as the calibration curve and the real sample analysis.

Thank you for the comment. The effectiveness of CoPPc/MWCNTs-COOH/GCE was evaluated by using amperometric i-t curves for simultaneous determination instead of DPV, so that DPV was replaced with amperometric i-t curves. The details are as follows (Page 13, line 447):

In this study, the effectiveness of CoPPc/MWCNTs-COOH/GCE was evaluated using amperometric i-t curves for the simultaneous determination of ascorbic acid (AA), uric acid (UA), dopamine (Dop), Sodium chloride (SC), Glucose (Glu) and hydrogen peroxide (HP) in phosphate-buffered saline (pH 7.0).

11) Table 1. Determination of lactic acid in rice wine samples.

“Hospital” should be replaced by “reference method”

Information about this reference method must be also provided.

Thank you for the comment. The “Hospital” was already replaced by “reference method”, and the reference method was national standard method GB 1886.173-2016. The details were as follows (Page 14, line 472-473, line 466):

Table 2. Determination of lactic acid in rice wine samples.

rice wine

samples

Lactic acid concentration (mM)

Error (%)

Recovery (%)

Reference method

Sensor

1

0.89

0.93

4.49%

101.9%

2

1.67

1.60

-4.19%

100.1%

3

1.24

1.31

5.65%

99.6%

4

0.68

0.72

5.88%

103.6%

Page 14, line 466: …the values obtained by rice wine company based on the standard method [50], …

References

  1. GB 1886.173-2016, Food additive - Lactic acid.

12) The samples were diluted by a factor of 81 (0.5 ml in 40 ml). The concentration is found in the range 0.7-1.6 mmol/L, meaning that the concentration after the dilution was 10-20 µmol/L. How the sensor performs at these concentrations of lactic acid? Figure 5 shows the graph starting from 24 µmol/L.

Thank you for the comment. The explanation was as follows:

The concentration of lactic acid starts from 10µmol/L, and the sensor performs well in the concentration from 10 to 240 μmol/L. The linear relationship between cathodic peak current and concentration of lactic acid could be calculated according to the following regression equation: ip (μA) = 0.1379c(μmol/L) + 0.0572, which the correlation coefficient could be calculated as 0.9988, and the LOD of 1×10-6 mol/L could be estimated when the SNR was 3.

 13) Was the method in Section 3.7 the same as in Section 3.5 and the figure with the calibration curve (Figure 5).

Thank you for the comment. The method used in the section 3.7 was added into the text, and details were as follows (Page 14, line 462):

… the linear sweep voltammetry (LSV) method with the optimal conditions …

Reviewer 3 Report

This manuscript presented a cobalt poly-phthalocyanine/carboxylic multi-walled carbon nanotube nanocomposites (CoPPc/MWCNTs-COOH) for lactic acid sensing in rice wine. The nanocomposite was systematically characterized, and the sensing performances towards lactic acid were evaluated. I recommend a major revision of this manuscript. Comments are as follows:

  1. The equation of the nanocomposite process should not appear in the abstract.
  2. The Section title of 3.5 is “Amperometric Determination of Lactic Acid”. Generally speaking, Amperometric Determination use a fixed potential to measure the stepwise increase in catalytic current. In fact, the linear sweep voltammetry method was used in the manuscript. So, I think the Section title should be changed.
  3. In Figure 6, there are two figures, (a) and (b), however, the figure caption only described the content of figure 6(b).
  4. Still figure 6(b), the current curve seems to be a Chronoamperometric response with a fixed potential, but the author described in their manuscript as: In the study, the effectiveness of CoPPc/MWCNTs-COOH/GCE was evaluated by using “DPV” for simultaneous determination of ascorbic acid (AA), uric acid (UA), ……. “DPV” or “Chronoamperometric current response”? the author should present their experiment result clearly.
  5. In real samples analysis, the sample is “rice wine”, why the sensor results were in good agreement with the values obtained by the “local hospital”? Direct describe the standard detection method might be better.

Author Response

Comments and Suggestions for Authors 3

This manuscript presented a cobalt poly-phthalocyanine/carboxylic multi-walled carbon nanotube nanocomposites (CoPPc/MWCNTs-COOH) for lactic acid sensing in rice wine. The nanocomposite was systematically characterized, and the sensing performances towards lactic acid were evaluated. I recommend a major revision of this manuscript. Comments are as follows:

1) The equation of the nanocomposite process should not appear in the abstract.

Thank you for the comment. The equation was deleted.

2) The Section title of 3.5 is “Amperometric Determination of Lactic Acid”. Generally speaking, Amperometric Determination use a fixed potential to measure the stepwise increase in catalytic current. In fact, the linear sweep voltammetry method was used in the manuscript. So, I think the Section title should be changed.

Thank you for the comment. The title of 3.5 was already changed, and the details were as follows (Page 12, line 408):

3.5. Determination of Lactic Acid Concentration

3) In Figure 6, there are two figures, (a) and (b), however, the figure caption only described the content of figure 6(b).

Thank you for the comment. The caption of Figure 6 was improved, and the details were as follows (Page 14, Line 458 - 459):

Figure 6. Stability test (a) and interference test(b) of the fabricated lactic acid sensor in a 100-fold concentration of Na+ and Cl-, 10-fold concentration of SC, AA, UA and Dop, and in the same concentration of Glu and HP.

4) Still figure 6(b), the current curve seems to be a Chronoamperometric response with a fixed potential, but the author described in their manuscript as: In the study, the effectiveness of CoPPc/MWCNTs-COOH/GCE was evaluated by using “DPV” for simultaneous determination of ascorbic acid (AA), uric acid (UA), ……. “DPV” or “Chronoamperometric current response”? the author should present their experiment result clearly.

Thank you for the comment. The simultaneous determination was evaluated by amperometric i-t curves but not DPV, so that “DPV” in Line 463 was replaced with “amperometric i-t curves”. The details were as follows (Page 13, line 447 - 451):

In this study, the effectiveness of CoPPc/MWCNTs-COOH/GCE was evaluated using amperometric i-t curves for the simultaneous determination of ascorbic acid (AA), uric acid (UA), dopamine (Dop), Sodium chloride (SC), Glucose (Glu) and hydrogen peroxide (HP) in phosphate-buffered saline (pH 7.0).

5) In real samples analysis, the sample is “rice wine”, why the sensor results were in good agreement with the values obtained by the “local hospital”? Direct describe the standard detection method might be better.

Thank you for the comment. The improper description was already corrected, and the content about “local hospital” was deleted, the details were as follows (Page 14, line 464 - 466):

It could be seen that the lactic acid concentrations tested by the as-prepared sensors were in good agreement with the values obtained by the rice wine Company based on the standard method [50], …

References

  1. GB 1886.173-2016, Food additive - Lactic acid.

Round 2

Reviewer 2 Report

The authors addressed most of my remarks. However, a few corrections are still needed

  1. As mentioned in my previous revision, Scheme 1 incorrectly depicts a peculiar pentameric fused Co phthalocyanine. There should be a monomeric Co phthalocyanine, which is used in the present work. The appearance of this type of pentamer is impossible for the given Pc, and not justified by the conditions, data or the previous literature.   
  2. The authors specified in the corrected manuscript that the current for the lactic acid signal is cathodic. The minus sign for the current seems to be missing in multiple figures  (4A, 4C , 6B).
  3. What is the potential applied in 6B, it should be specified for clarity. If stirring was used during the measurements this also should be mentioned.
  1. The authors specified that the observed electrochemical process related to lactic acid is cathodic, which is rather surprising taking into account the literature (for example, ref 43, https://www.mdpi.com/2227-9040/9/4/74) that refers to an anodic process. Since the cathodic process is rather unexpected, the authors should provide relevant references to the lactic acid electrochemical reduction or explain the chemistry behind it.
  1. In the response, the authors mentioned that MWCNT/GCE does not give any effect when lactic acid is present in the solution. Would it be possible add this curve in Fig 3A? It is a more important and relevant blank control experiment compared to a bare GC electrode currently used in Fig 3A.

Author Response

Thank you for the review, the detailed responses were added in the attachment, thank you.

Reviewer 3 Report

The author have revised the manuscript according to reviewer's comments.

Author Response

Thank you for the review